# Neural heterogeneity promotes robust learning

Nicolas Perez-Nieves [1✉], Vincent C. H. Leung [1], Pier Luigi Dragotti[1] & Dan F. M. Goodman [1✉]

The brain is a hugely diverse, heterogeneous structure. Whether or not heterogeneity at the neural level plays a functional role remains unclear, and has been relatively little explored in models which are often highly homogeneous. We compared the performance of spiking neural networks trained to carry out tasks of real-world difficulty, with varying degrees of heterogeneity, and found that heterogeneity substantially improved task performance. Learning with heterogeneity was more stable and robust, particularly for tasks with a rich temporal structure. In addition, the distribution of neuronal parameters in the trained networks is similar to those observed experimentally. We suggest that the heterogeneity observed in the brain may be more than just the byproduct of noisy processes, but rather may serve an active and important role in allowing animals to learn in changing environments.

[1] Department of Electrical and Electronic Engineering, Imperial College London, London SW7 2AZ, UK. ✉email: nicolas.perez14@imperial.ac.uk; d.goodman@imperial.ac.uk

The brain is known to be deeply heterogeneous at all scales[1], but it is still not known whether this heterogeneity plays an important functional role or if it is just a byproduct of noisy developmental processes and contingent evolutionary history. A number of hypothetical roles have been suggested (reviewed in ref. [2]), including in efficient coding[3–9], reliability[10], working memory[11], and functional specialisation[12]. However, previous studies have largely used simplified tasks or networks, and it remains unknown whether or not heterogeneity can help animals solve complex information processing tasks in natural environments. Recent work has allowed us, for the first time, to train biologically realistic spiking neural networks to carry out these tasks at a high level of performance, using methods derived from machine learning. We used two different learning models[13,14] to investigate the effect of introducing heterogeneity in the time scales of neurons when performing tasks with realistic and complex temporal structure. We found that it improves the overall performance, makes learning more stable and robust, and that the network learns neural parameter distributions that match experimental observations, suggesting that the heterogeneity observed in the brain may be a vital component of its ability to adapt to new environments.

## Results

### Time scale heterogeneity improves learning on tasks with rich temporal structure.

We investigated the role of neural heterogeneity in task performance by training recurrent spiking neural networks to classify visual and auditory stimuli with varying degrees of temporal structure. The model used three layers of spiking neurons: an input layer, a recurrently connected layer, and a readout layer used to generate predictions (Fig. 1A), a widely used minimal architecture (e.g. Maass et al.[15], Neftci et al.[14]). Heterogeneity was introduced by giving each neuron an individual membrane and synaptic time constant. We compared four different conditions: initial values could be either homogeneous or heterogeneous, and training could be either standard or heterogeneous (Fig. 1B). In more detail, time constants were either initialised with a single value (homogeneous initialisation), or randomly according to a gamma distribution (heterogeneous initialisation). In both types of training, the parameters of the models were optimised using surrogate gradient descent[14]. Synaptic weights were trainable in both standard and heterogeneous training regimes, while time constants were either held fixed at their initial values in the standard training regime, or could be modified in the heterogeneous training regime.

We used five different datasets with varying degrees of temporal structure. Neuromorphic MNIST (N-MNIST)[16], Fashion-MNIST (F-MNIST)[17], and the DVS128 Gesture dataset[18] feature visual stimuli, while the Spiking Heidelberg Digits (SHD) and Spiking Speech Commands (SSC) datasets[19] are auditory. N-MNIST and DVS128 use a neuromorphic vision sensor to generate spiking activity, by moving the sensor with a static visual image of handwritten digits (N-MNIST) or by recording humans making hand gestures (DVS128). F-MNIST is a dataset of static images that are widely used in machine learning, which we converted into spike times by treating the image intensities as input currents to model neurons, so that higher intensity pixels would lead to earlier spikes, and lower intensity to later spikes. Both SHD and SSC use the same detailed model of the activity of bushy cells in the cochlear nucleus developed in Cramer et al.[19], in response to spoken digits (SHD) or commands (SSC). Their model consists of standard components from the auditory modelling literature, including a hydrodynamic basilar membrane, transmitter pool and inhomogeneous Poisson process hair cell, and leaky integrate-and-fire

bushy cell. Of these datasets, N-MNIST and F-MNIST have minimal temporal structure, as they are generated from static images. DVS128 has some temporal structure as it is recorded in motion, but it is possible to perform well at this task by discarding the temporal information. The auditory tasks SHD and SSC by contrast have very rich temporal structure. In all cases, we used the train/test split suggested by the corresponding dataset authors to compare our performance with previous baselines fairly.

We found that heterogeneity in time constants had a profound impact on performance on those training datasets where information was encoded in the precise timing of input spikes (Table 1 and Fig. 2A). On the most temporally complex auditory tasks, accuracy improved by a factor of around 15–20%, while for the least temporally complex task N-MNIST, we saw no improvement at all. For the gesture dataset DVS128, we can identify the source of the (intermediate) improvement as the heterogeneous models being better able to distinguish between spatially similar but temporally different gestures, such as clockwise and anticlockwise versions of the same gesture (Supplementary Fig. 7A, B). This suggests that we might see greater improvements for a richer version of this dataset in which temporal structure was more important.

We verified that our results were due to heterogeneity and not simply to a better tuning of time constants in two ways. Firstly, we performed a grid search across all homogeneous time constants for the SHD dataset and used the best values for our comparison. Secondly, we observe that the distribution of time constants after training is very similar and heterogeneous regardless of whether it was initialised with a homogeneous or heterogeneous distribution (Fig. 2B), indicating that the heterogeneous distribution is optimal.

Introducing heterogeneity allows for a large increase in performance at the cost of only a very small increase in the number of parameters (0.23% for SHD, because we have added some neuron-specific parameters, but not added any synapse-specific ones, and the vast majority of parameters are synaptic weights), and without using any additional neurons or synapses. Heterogeneity is therefore a metabolically efficient strategy. It is also a computationally efficient strategy of interest to neuromorphic computing, because adding heterogeneous time constants to the model adds $O(n)$ to memory use and computation time when simulating the model, while adding more neurons adds $O(n^2)$ (because the model is fully connected, so the number of synaptic weights is proportional to the square of the number of neurons).

Note that it is possible to obtain better performance using a larger number of neurons. For example, Cramer et al.[19] obtained a performance of 83.2% on the SHD dataset without heterogeneity using 1024 neurons and data augmentation techniques, whereas we obtained 82.7% using 128 neurons and no data augmentation. We focus on smaller networks here for two reasons. Firstly, we wanted to systematically investigate the effect of different training regimes, and the current limitations of surrogate gradient descent mean that each training session takes several days. Secondly, with larger numbers of neurons, performance even without heterogeneity can in some cases approach the ceiling on these tasks (which are still simple in comparison to those faced by animals in real environments), making it more difficult to see the effect of different architectures. In the case of the SSC dataset, however, even with this limitation to small networks, heterogeneity confers such an advantage that our results are state of the art (for spiking neural networks) by a large margin.

We also tested the effect of introducing heterogeneity of other neuron parameters, such as the firing threshold and reset potential, but found that it made no appreciable difference. This

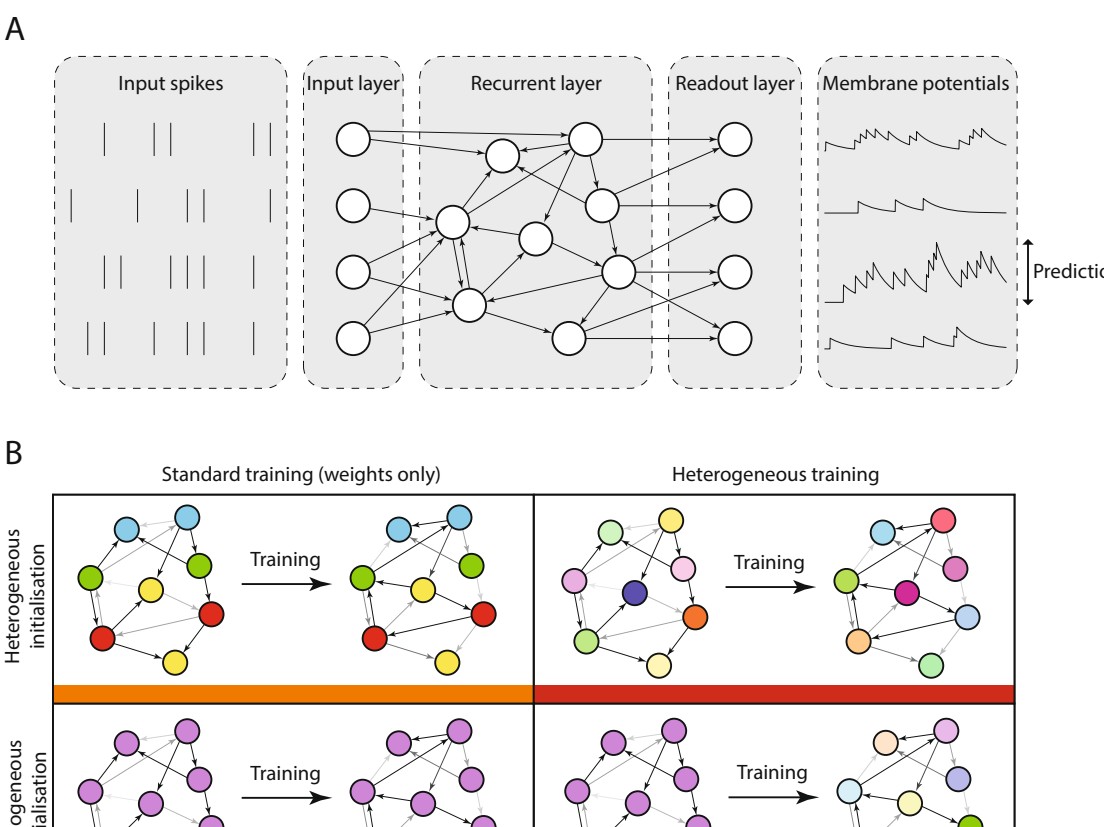

**Fig. 1 Diagram of network architecture and training configurations. A** Model architecture. A layer of input neurons emits spike trains into a recurrently connected layer of spiking neurons which is followed by a readout layer. **B** Configurations. Training can be either standard (only the synaptic weights are learned) or heterogeneous (the synaptic weights and membrane and synaptic time constants are learned). The initialisation can be homogeneous (all synaptic and membrane time constants are initialised to the same value) or heterogeneous (synaptic and membrane time constants are randomly initialised for each neuron by sampling them from a given probability distribution).

**Table 1 Testing accuracy percentage over different datasets and training methods.**

| Initialisation | Training | N-MNIST | F-MNIST | DVS128 | SHD | SSC |
|---|---|---|---|---|---|---|
| Homog. | Standard | 97.4 ± 0.0 | 80.1 ± 7.4 | 76.9 ± 0.8 | 71.7 ± 1.0 | 49.7 ± 0.4 |
| Heterog. | Standard | 97.5 ± 0.0 | 87.9 ± 0.1 | 79.5 ± 1.0 | 73.7 ± 1.1 | 53.7 ± 0.7 |
| Homog | Heterog. | 96.6 ± 0.2 | 79.7 ± 7.4 | 81.2 ± 0.8 | 82.7 ± 0.8 | 56.6 ± 0.7 |
| Heterog. | Heterog. | 97.3 ± 0.1 | 87.5 ± 0.1 | 82.1 ± 0.8 | 81.7 ± 0.8 | 60.1 ± 0.7 |
| Chance level | | 10.0 | 10.0 | 10.0 | 5.0 | 2.9 |

Effect of initialisation and training configuration on performance, on datasets of increasing temporal complexity. Initialisation can be homogeneous (all time constants the same) or heterogeneous (random initialisation), and training can be standard (only synaptic weights learned) or heterogeneous (time constants can also be learned). N-MNIST and F-MNIST are static image datasets with little temporal structure, DVS128 is video gestures, and SHD and SSC are temporally complex auditory datasets.

was because for our model, changing these is almost equivalent to a simple scaling of the membrane potential variable. By contrast, Bellec et al.[24] found that introducing an adaptive threshold did improve performance, presumably because it allows for much richer temporal dynamics (in line with earlier findings[25]).

**Predicted time constant distributions match experimental data**. In all our tasks, the distributions of time constants after training approximately but not exactly fit a log-normal or gamma distribution (with different parameters for each task). They are also consistent across different training runs (Supplementary Figs. 1 and 2) and initial distributions (Supplementary Figs. 8 and 9), suggesting that the learned distributions may be optimal.

Using publicly available datasets including time constants recorded in large numbers of neurons in different animals and brain regions[20–23], we found similar distributions to those we predicted (Fig. 2C). This is striking, as our model is optimised for a single, relatively simple task, while animals in real environments face a range of tasks, each of which is more difficult than our model tasks. The parameters for the experimentally observed distributions are different for each animal and region, just as for different tasks in our simulations. Interestingly, the distribution parameters are also different for each cell type in the experimental data, a feature not replicated in our simulations as all cells are identical. This suggests that introducing further diversity in terms of different cell types may lead to even better performance.

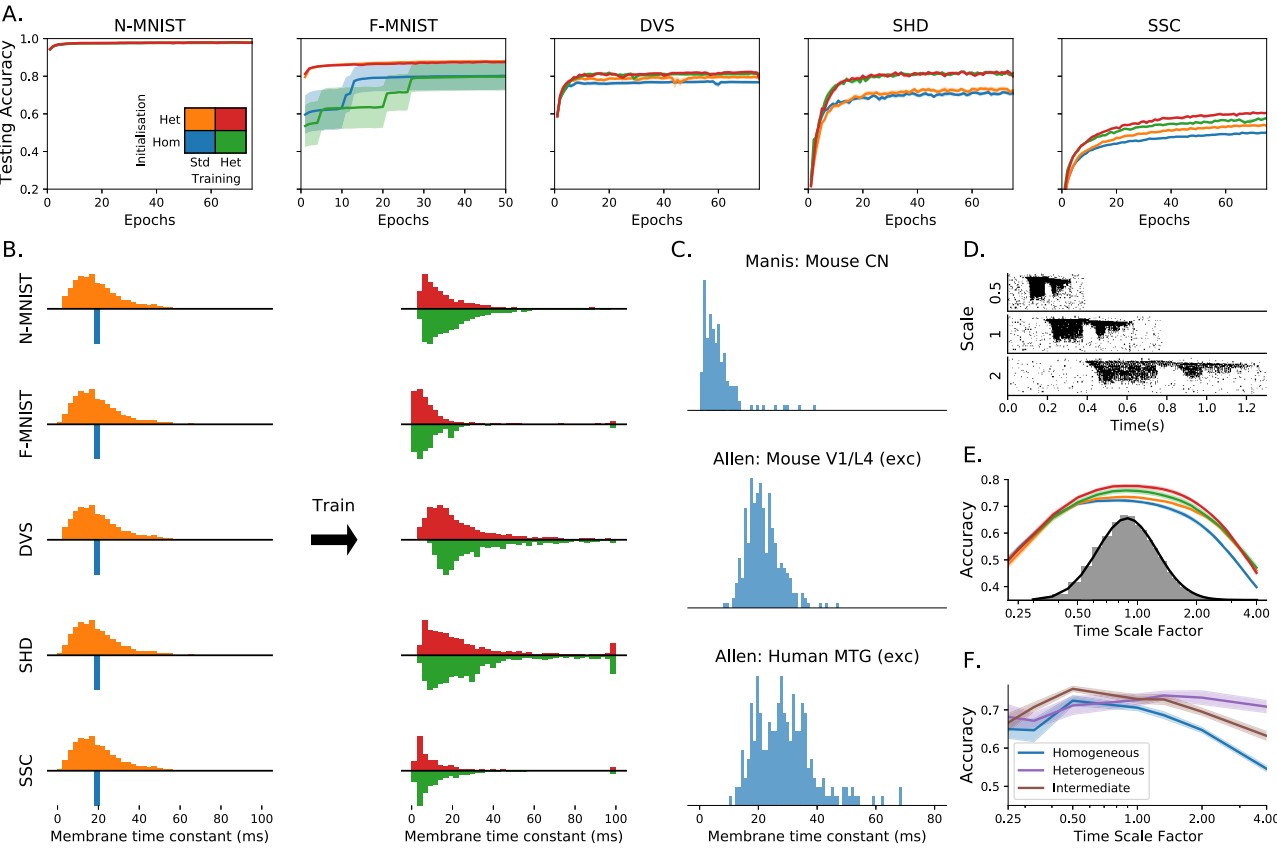

**Fig. 2 Impact of training configuration and temporal structure of the dataset on the testing accuracy, membrane time constant distributions and performance when training at different time scales. A** Improvements in accuracy in testing data, for datasets with low (N-MNIST, F-MNIST), intermediate (DVS) and high (SHD) temporal complexity. Shaded areas correspond to the standard error of the mean over 10 trials (which in some cases is too small to be visible). Initialisation can be homogeneous (blue/green) or heterogeneous (orange/red), and training can be standard, weights only (blue/ orange) or heterogeneous including time constants (green/red). Heterogeneous configurations achieve a better test accuracy on the more temporally complex datasets. Heterogeneous initialisation also results in a more stable and robust training trajectory for F-MNIST, leading to a better performance overall. **B** Membrane time constant distributions before (left) and after (right) training for each dataset. Histograms above the axis represent heterogeneous initialisation, and below the axis homogeneous initialisation. In the case of standard training (weights only), the initial distribution (left) is the same as the final distribution of time constants after training. **C** Experimentally observed distributions of time constants for (top to bottom): mouse cochlear nucleus, multiple cell types (172 cells)[20,21]; mouse V1 layer 4, spiny (putatively excitatory) cells (164 cells)[22,23]; human middle temporal gyrus, spiny cells (236 cells)[22,23]. **D** Raster plot of input spikes from a single sample of the SHD dataset (spoken digits) at three different time scales. **E** Accuracy on the SHD dataset after training on a variety of time scales (randomly selected from the grey distribution) for the four configurations described in **A**. **F** Accuracy on the SHD dataset when the initial distribution of time constants is tuned for time scale 1.0, but the training and testing is done at different time scales.

**Heterogeneity improves generalisation: speech learning across time scales**. Sensory signals such as speech and motion can be recognised across a range of speeds by humans and animals. We tested the role of heterogeneity in learning a circuit that can function across a wide range of speeds. We augmented the SHD spoken digits datasets to include faster or slower versions of the samples, multiplying all spike times by a temporal scale as an extremely simplified model that captures a part of the difficulty of this task (Fig. 2D). During training, temporal scales were randomly selected from a distribution roughly matching human syllabic rate distributions[26]. The heterogeneous configurations performed as well or better at all time scales (Fig. 2E), and in particular were able to generalise better to time scales outside the training distribution (e.g. an accuracy of 47% close to a time scale of 4, around 7% higher than the homogeneous network, where chance performance would be 5%). Heterogeneous initialisation alone was sufficient to achieve this better generalisation performance, while time constant training improved the peak performance but gave no additional ability to generalise.

**Heterogeneity improves robustness against mistuned learning**. We tested the hypothesis that heterogeneity can provide robustness with two experiments where the hyperparameters were mistuned, that is where the initial distributions and learning parameters were chosen to give the best performance for one distribution, but the actual training and testing is done on a different distribution.

In the first experiment, we took the augmented SHD spoken digits dataset from the previous section, selected the hyperparameters to give the best performance at a time scale of 1, but then trained and tested the network at a different time scale (Fig. 2F). We used training of weights only, since allowing retraining of time constants lets the network cancel out the effect of changing the time scale. With a homogeneous or narrow heterogeneous initial distribution (blue line in Fig. 2F), performance falls off for time scales far from the optimal one, particularly for larger time scales (slower stimuli). However, a wide heterogeneous initial distribution (purple line in Fig. 2F) allows for good performance across all time scales tested, at the cost of slightly lower peak

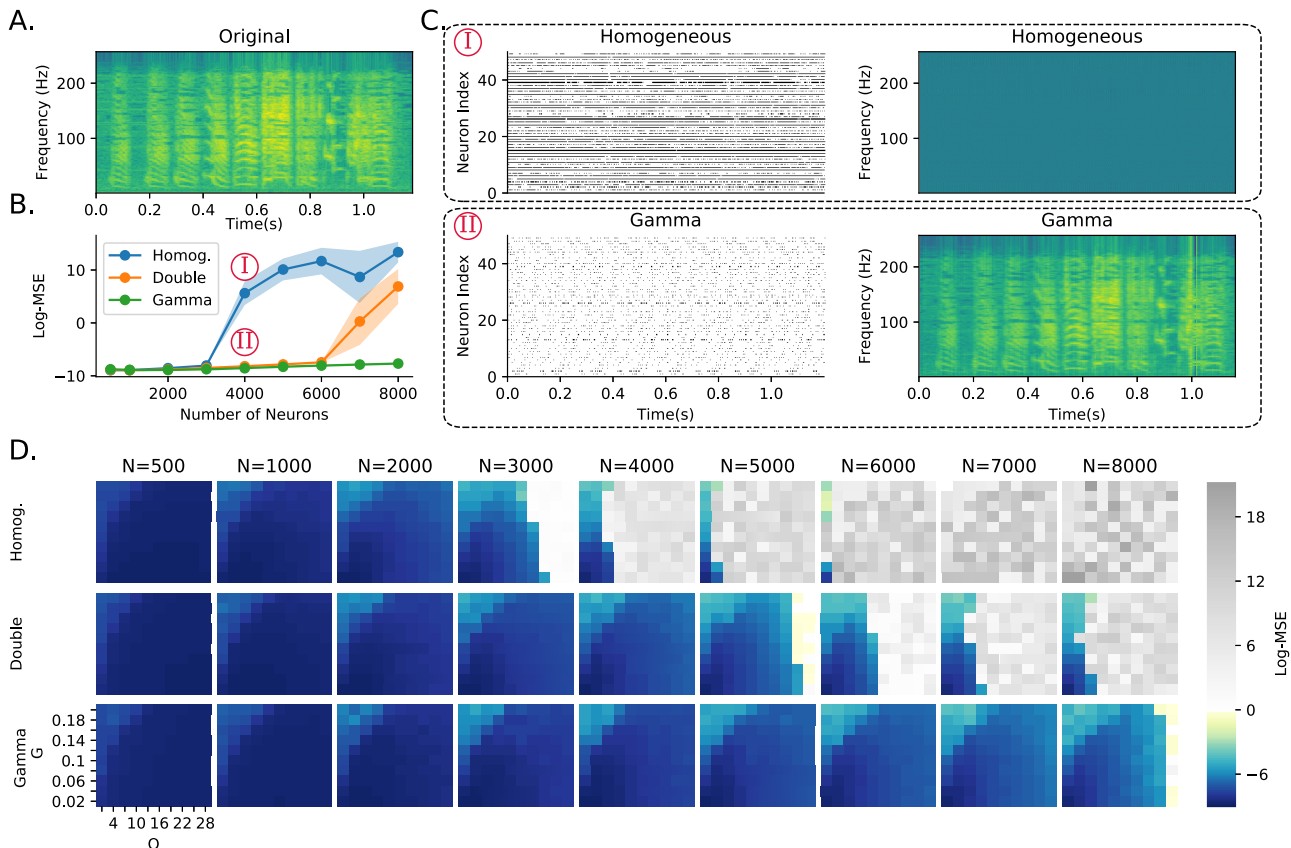

**Fig. 3 Robustness to learning hyperparameter mistuning. A** Spectrogram of a zebra finch call. The network has to learn to reproduce this spectrogram, chosen for its spectrotemporal complexity. **B** Error for three networks at different network sizes (hyperparameters G and Q were chosen to optimise performance at N = 1000 neurons as given in Table 4). Networks are fully homogeneous (Homog); intermediate, where each neuron is randomly assigned slow or fast dynamics (Double); or fully heterogeneous, where each neuron has a random time constant drawn from a gamma distribution (Gamma). **C** Raster plots of 50 neurons randomly chosen, and reconstructed spectrograms under fully homogeneous and fully heterogeneous (Gamma) conditions for N = 4000 neurons as indicated in **B**. **D** Reconstruction error. Each row is one of the three configurations shown as lines in **B**. Each column is a network size. The axes of each image give the learning hyperparameters (G and Q). Grey pixels correspond to log mean square error above 0, corresponding to a complete failure to reconstruct the spectrogram. The larger the coloured region, the more robust the network is, and less tuning is required.

performance at the best time scale. We tested whether or not this was solely due to the presence of a few neurons with long time constants by introducing an intermediate distribution where the majority of time constants were the same as the homogeneous case, with a small minority of much longer time constants (brown line in Fig. 2F). The performance of this distribution was intermediate between the homogeneous and heterogeneous distributions, a pattern that is repeated in our second experiment below.

In the second experiment, we switched to a very different learning paradigm, FORCE training of spiking neural networks[13] to replay a learned signal, in this case, a recording of a zebra finch call (Fig. 3A; from Blättler and Hahnloser[27]). This method does not allow for heterogeneous training (tuning time constants), so we only tested the role of untrained heterogeneous neuron parameters. We tested three configurations: fully homogeneous (single 20 ms time constant as in the original paper); intermediate (each neuron randomly assigned a fixed fast 20 ms or slow 100 ms time constant); or fully heterogeneous (each neuron randomly assigned a time constant drawn from a gamma distribution).

Nicola and Clopath[13] showed that network performance is highly dependent on two hyperparameters (G and Q in their paper). We, therefore, tuned these hyperparameters for a network of a fixed size (N = 1000 neurons) and ran the training and testing for networks of different sizes (Fig. 3B). As the network size started to diverge, the homogeneous network began to make

large errors, while the fully heterogeneous network was able to give low errors for all network sizes. The intermediate network was able to function well across a wider range of network sizes than the fully homogeneous network, but still eventually failed for the largest network sizes. At these large network sizes, the homogeneous network neurons fire at or close to their maximal rate and training cannot take place, leading to poor performance (Fig. 3C). The robustness of the heterogeneous version of the network can be measured by the area of the hyperparameter space that leads to good performance (Fig. 3D). Adding partial or full heterogeneity leads to an improvement in learning for all points in the hyperparameter space, again suggesting that it can improve the robustness of learning in a wide range of situations.

## Discussion

We trained spiking neural networks at difficult classification tasks, either forcing all neuron time constants to be the same (homogeneous) or allowing them to be different (heterogeneous). We found that introducing heterogeneity improved the overall performance across a range of tasks and training methods, but particularly so on tasks with richer intrinsic temporal structure. Learning was more robust, for heterogeneous networks, in that the networks were able to learn across a range of different environments, and when the hyperparameters of learning were mistuned. When the learning rule was allowed to tune the time

constants as well as synaptic weights, a consistent distribution of time constants was found, akin to a log-normal or gamma distribution, and this qualitatively matched time constants measured in experimental data. Our model is consistent with the two possibilities that these time constant distributions are learned during an individuals lifetime, or that they are found as a result of an evolutionary process.

We conclude from this that neural heterogeneity is a metabolically efficient strategy for the brain. Heterogeneous networks have no additional cost in terms of the number of neurons or synapses, and perform as well as homogeneous networks which have an order of magnitude more neurons. This gain also extends to neuromorphic computing systems, as adding heterogeneity to the neuron model adds an additional time and memory cost of only $O(n)$, while adding more neurons has a cost of $O(n^2)$ (because all neurons are connected). Further, in some neuromorphic systems like BrainScaleS this heterogeneity is already present as part of the manufacturing process[28]. In addition to their overall performance being better, heterogeneous networks are more robust and able to learn across a wider range of environments, which is clearly ethologically advantageous. Again, this has a corresponding benefit to neuromorphic computing and potentially machine learning more generally, in that it reduces the cost of hyperparameter tuning, which is often one of the largest costs for developing these models.

We now turn to the matter of the extent of time constant tuning in real nervous systems. It could be the case that the heterogeneous distribution of time constants observed in different animals and brain regions (Fig. 2C) is simply a product of noisy developmental processes. We cannot rule this out. However, our results show that the distributions observed experimentally closely match the optimal ones found by simulations, and this optimal distribution confers a substantial computational advantage. It, therefore, seems likely that the brain makes use of this advantage. We found that any degree of heterogeneity improves performance, but that the best performance could be found by tuning the distribution of time constants to match the task. Without a more detailed model of particular brain regions in specific animals, and the real-world tasks they solve, it is difficult to conclude whether or not the precise distributions observed experimentally are tuned to those tasks or not. Further, having a less precisely tuned distribution may lead to greater robustness in uncertain environments.

A number of studies have used heterogeneous or tunable time constants[29–31], but these have generally been focussed on maximising performance for neuromorphic applications, and not considering the potential role in real nervous systems. In particular, we have shown that: heterogeneity is particularly important for the type of temporally complex tasks faced in real environments, as compared to the static ones often considered in machine learning; heterogeneity confers robustness allowing for learning in a wide range of environments; optimal distributions of time constants are consistent across training runs and match experimental data; and that our results are not specific to a particular task or training method.

The methods used here are very computationally demanding, and this has limited us to investigating very small networks (hundreds of neurons when using surrogate gradient descent and thousands using FORCE training). Indeed, we estimate that in the preparation of this paper, we used approximately 2 years of GPU computing. Finding new algorithms to allow us to scale these methods to larger networks will be a critical task for the field.

Finally, it would be interesting to see to what extent different forms of heterogeneity confers other advantages, such as spatial as well as temporal heterogeneity. We observed that in the brain, different cell types have different stereotyped distributions of time

constants, and it would be interesting to extend our methods to networks with multiple cell types, including more biophysically detailed cell models.

Our computational results show a compelling advantage for heterogeneity, and this makes intuitive sense. Having heterogeneous time constants in a layer allows the network to integrate incoming spikes at different time scales, corresponding to shorter or longer memory traces, thus allowing the readout layer to capture information at several scales and represent a richer set of functions. It would be very valuable to extend this line of thought and find a rigorous theoretical explanation of the advantage of heterogeneity.

## Methods

**Neuron and synaptic models.** We use the Leaky Integrate and Fire (LIF) neuron model in all our simulations. In this model, the membrane potential of the $i$th neuron in the $l$th layer $U_i^{(l)}(t)$ varies over time following Eq. (1).

$$\tau_{\mathrm{m}} \dot{U}_i^{(l)}(t) = -(U_i^{(l)}(t) - U_0) + I_i^{(l)}(t) \tag{1}$$

Here, $\tau_{\mathrm{m}}$ is the membrane time constant, $U_0$ is the resting potential and $I_i^{(l)}$ is the input current. When the membrane potential reaches the threshold value $U_{\mathrm{th}}$ a spike is emitted, $U_i(t)$ resets to the reset potential $U_{\mathrm{r}}$ and then enters a refractory period that lasts $t_{\mathrm{ref}}$ seconds where the neuron cannot spike.

Spikes emitted by the $j$th neuron in layer $l-1$ at a finite set of times $\{t_j^{(k)}\}$ can be formalised as a spike train $S_j^{(l-1)}(t)$ defined as in Eq. (2)

$$S_j^{(l-1)}(t) = \sum_k \delta(t - t_j^{(k)}) \tag{2}$$

The input current $I_i^{(l)}$ is obtained from the spike trains of all presynaptic neurons $j$ connected to neuron $i$ following Eq. (3)

$$\tau_{\mathrm{s}} \dot{I}_i^{(l)}(t) = -I_i^{(l)}(t) + \sum_j W_{ij}^{(l)} S_j^{(l-1)}(t) + \sum_j V_{ij}^{(l)} S_j^{(l)}(t) \tag{3}$$

Here $\tau_{\mathrm{s}}$ is the synaptic time constant, $W_{ij}^{(l)}$ is the feed-forward synaptic weight from neuron $j$ in layer $l-1$ to neuron $i$ in layer $l$ and $V_{ij}^{(l)}$ is the recurrent weight from neuron $j$ in layer $l$ to neuron $i$ in layer $l$.

Thus, a LIF neuron is fully defined by six parameters $\tau_{\mathrm{m}}, \tau_{\mathrm{s}}, U_{\mathrm{th}}, U_0, U_{\mathrm{r}}, t_{\mathrm{ref}}$ plus its synaptic weights $W_{ij}^{(l)}$ and $V_{ij}^{(l)}$. We refer to these as the neuron parameters and weights, respectively.

Since we are considering the cases where these parameters may be different for each neuron in the population we should actually refer to $\tau_{\mathrm{m},i}, \tau_{\mathrm{s},i}, U_{\mathrm{th},i}, U_{0,i}, U_{\mathrm{r},i}, t_{\mathrm{ref},i}$. However, for notational simplicity, we will drop the $i$ subscript and it will be assumed that these parameters can be different for each neuron in a population.

**Neural and synaptic model discretisation.** In order to implement the LIF model in a computer, it is necessary to discretise it. We follow the discretisation in Cramer et al.[19]. Here we use square brackets to index variables changing in a discrete time setting, so that $t$ refers to a continuous value when written $f(t)$ or an integer when written $f[t]$. We discretise time into multiples of a small-time step $\Delta t$, so that spikes can only happen at multiples of $\Delta t$. With this discretisation, we can approximately solve Eq. (3) to give

$$I_i^{(l)}(t + \Delta t) = I_i^{(l)}[t + 1] = \alpha I_i^{(l)}[t] + \sum_j W_{ij} S_j^{(l-1)}[t] + \sum_j V_{ij} S_j^{(l)}[t], \tag{4}$$

with $\alpha = \exp(-\Delta t / \tau_{\mathrm{s}})$. Similarly, Eq. (1) becomes

$$U_i^{(l)}(t + \Delta t) = U_i^{(l)}[t + 1] = \beta(U_i^{(l)}[t] - U_0) + U_0 + (1 - \beta)I_i^{(l)}[t] - (U_{th} - U_r)S_i^{(l)}[t], \tag{5}$$

with $\beta = \exp(-\Delta t / \tau_{\mathrm{m}})$. Finally, the spiking mechanism

$$S_i^{(l)}[t] = \begin{cases} 1 & \text{if } U_i^{(l)}[t] - U_{\mathrm{th}} \geq 0 \\ 0 & \text{if } U_i^{(l)}[t] - U_{\mathrm{th}} < 0 \end{cases} \tag{6}$$

Notice how the last term in Eq. (5) introduces the membrane potential resetting. This would only work if we assume that the neuron potential at spiking time was exactly equal to $U_{\mathrm{th}}$. This may not necessarily be the case since a membrane potential update that crossed the threshold may result in $U_i^{(l)} > U_{\mathrm{th}}$ and then the resetting mechanism will not set the membrane potential to $U_{\mathrm{r}}$. However, we found that this has a negligible effect in our simulations.

**Surrogate gradient descent training.** With the discretisation introduced in the previous section, a spiking layer consists of three cascaded sub-layers: current from Eq. (4), membrane from Eq. (5) and spike from Eq. (6). The current and membrane sub-layers have access to their previous state and thus, they can be seen as a particular case of recurrent neural network (RNN). Note that while each neuron is

**Table 2 Parameter initialisation for the different configurations.**

| Parameter | HomInit | HetInit | Description |
|---|---|---|---|
| $\tau_m$ | $\bar{\tau}_m$ | $\Gamma(3, \bar{\tau}_m/3)$ | Membrane time constant |
| $\tau_s$ | $\bar{\tau}_s$ | $\Gamma(3, \bar{\tau}_s/3)$ | Synaptic time constant |
| $U_{th}$ | $U_{th}$ | $\mathcal{U}(0.5, 1.5)$ | Membrane threshold |
| $U_0$ | $U_0$ | $\mathcal{U}(-0.5, 0.5)$ | Resting potential |
| $U_r$ | $U_r$ | $\mathcal{U}(-0.5, 0.5)$ | Reset potential |

**Table 3 Surrogate gradient descent network parameters.**

| Parameter | Value | Description |
|---|---|---|
| $\Delta t$ | 0.5 ms | Simulation time step |
| $\bar{\tau}_m$ | 20 ms | Mean membrane time constant |
| $\bar{\tau}_s$ | 10 ms | Mean synaptic time constant |
| $U_{th}$ | 1 V | Membrane threshold |
| $U_0$ | 0 mV | Resting potential |
| $U_r$ | 0 mV | Reset potential |
| $t_{ref}$ | 0 ms | Refractory time |
| $\rho$ | 100 | Surrogate steepness |

a recurrent unit since it has access to its own previous state, different neurons in the same spiking layer will only be connected if any of the non-diagonal elements of $\boldsymbol{V}^{(l)}$ is non-zero. In other words, all spiking neural networks (SNNs) built using this model are RNNs but not all SNNs are recurrent SNNs (RSNNs).

We can cascade $L$ spiking layers to conform a deep spiking neural network analogous to a conventional deep neural network and train it using gradient descent. However, as Eq. (6) is non-differentiable, we need to modify the backwards pass as in ref. [14] so that the backpropagation through time (BPTT) algorithm can be used to update the network parameters.

$$\sigma\left(U_i^{(l)}\right) = \frac{U_i^{(l)}}{1 + \rho|U_i^{(l)}|} \qquad (7)$$

This means that while in the forward pass the network follows a step function as in Eq. (6), in the backwards pass it follows a sigmoidal function described in Eq. (7), with a steepness set by $\rho$.

We can now use gradient descent to optimise the synaptic weights $\boldsymbol{W}^{(l)}$ and $\boldsymbol{V}^{(l)}$ as in conventional deep learning. We can also optimise the spiking neuron-specific parameters $U_{th}, U_0, U_r$ since they can be seen as bias terms. The time constants can also be indirectly optimised by training $\alpha$ and $\beta$, which can be seen as forgetting factors.

We apply a clipping function to $\alpha$ and $\beta$ after every update.

$$\text{clip}(x) = \begin{cases} e^{-1/3}, & \text{if } x < e^{-1/3} \\ 0.995, & \text{if } x > 0.995 \end{cases} \qquad (8)$$

In order to ensure stability, the forgetting factors have to be <1. Otherwise, the current and membrane potential would grow exponentially. Secondly to make the system causal these factors cannot be less than zero. This, however, would allow for arbitrarily small time constants and for a finite time resolution $\Delta t$ leads to constant firing and numerical instability. Thus, we constrain the time constants to be at least $3\Delta t$. This leads to the minimum value being $\exp(-1/3)$. As for the upper bound, we chose a maximum time constant of 100 ms (corresponding to $\exp(-\Delta t/100) = 0.995$) informed by the time constants reported in the NeuroElectro database[32], which shows that about 99.5% of the membrane time constants recorded from biological neurons have time constants below this value. The lower bound is also consistent with this database given our simulation time step of 0.5 ms. Since there is no data available on synaptic time constants, we chose the same range as for the membrane constant. We also set clipping limits for $U_{th}, U_0, U_r$ such that they are always between the ranges specified in Table 2.

There are several ways in which the neuron parameters may be trained. One possibility is to make all neurons in a layer share the same neuron parameters. That is, a single value of $U_{th}, U_0, U_r, \alpha, \beta$ is trained and shared by all neurons in a layer. Another possibility is to optimise each of these parameters in each neuron individually as we have done in our experiments. We also always trained the weight matrices $\boldsymbol{W}^{(l)}$ and $\boldsymbol{V}^{(l)}$. Training was done by using automatic differentiation on PyTorch[33] and Adam optimiser with a learning rate $10^{-3}$ and betas (0.9, 0.999).

In all surrogate gradient descent experiments, a single recurrent spiking layer with 128 neurons received all input spikes. This recurrent layer is followed by a feed-forward readout layer with $U_{th}$ set to infinity and with as many neurons as classes in the dataset.

For the loss function, we follow the *max-over-time loss* in ref. [19] to take the maximal membrane potential over the entire time in the readout layer, which was found to lead to the highest classification performance. We then take these potentials and compute the cross-entropy loss

$$\mathcal{L} = -\log\left(\frac{\exp\left(\arg\max_t U_{class}^{(L)}[t]\right)}{\sum_j \exp\left(\arg\max_t U_j^{(L)}[t]\right)}\right) \qquad (9)$$

where class corresponds to the readout neuron index of the correct label for a given sample. The loss is computed as the average of $N_{batch}$ training samples. This was repeated for a total of $N_{epochs}$.

In order to improve generalisation, we added noise to the input by adding spikes following a Poisson process with a rate of 1.2 Hz and deleting spikes with a probability of 0.001.

The parameters used for the network are given in Tables 2 and 3, unless otherwise specified. In Fig. 2F, we used a log-normal distribution for the initial values of the time constants in which we ensured the mode was the same as in the

Gamma distribution, we used for the other experiments (Table 2) but we scaled the standard deviation to be $f$ times that of the original Gamma distribution. We used $f = 4$ for the heterogeneous case.

For the intermediate case, all neurons were initialised as in the Homogeneous configuration except for 5% of them selected randomly that were given the largest time constant value allowed of 100 ms. The aim of this was to test whether all that was needed was some neurons with a longer memory. We chose 5% since the optimal distribution of trained time constants in the SHD shows that about this fraction of neurons reach this maximum value.

All states $I_i(l)[0]$ and $U_i(l)[0]$ are initialised to 0. For the weights $\boldsymbol{W}$ and $\boldsymbol{V}$, we independently sampled from a uniform distribution $\mathcal{U}(-k^{-1/2}, k^{-1/2})$, with $k$ being the number of afferent connections[34].

**FORCE training.** The FORCE method is used to train a network consisting of a single recurrent layer of LIF neurons as in ref. [13]. In particular, we followed the method used to learn to reproduce a songbird singing. In this method, there are no feed-forward weights and only the recurrent weights $V$ are trained. We can express these weights as

$$V = Gv + Q\eta\phi^T \qquad (10)$$

The first term in Eq. (10), namely $Gv$, remains static during training and it is initialised to set the network into chaotic spiking. The learned component of the weights $\phi^T \in \mathbb{R}^{K \times N}$ is updated using the Recursive Least Squares algorithm. The vector $\eta \in \mathbb{R}^{N \times K}$ serves as a decoder and it is static during learning. The constants $G$ and $Q$ govern the ratio between chaotic and learned weights.

With this definition of $V$, we can write the currents into the neurons as the sum $I[t] = I_G[t] + I_Q[t]$ (we dropped the layer $l$ superscript since we only have a single layer) where we define

$$I_G[t + 1] = \alpha I_G[t] + GvS[t] \qquad (11)$$

$$I_Q[t + 1] = Q\eta\phi^T r[t] \qquad (12)$$

$$r[t + 1] = \alpha r[t] + S[t] \qquad (13)$$

In order to stabilise the network dynamics, we add a High Dimensional Temporal Signal (HDTS) as in ref. [13]. This is an $M$ dimensional periodic signal $z[t]$. Given the HDTS period $T$, we split the interval $[0, T]$ into $M$ subintervals $I_m, m = 1, \ldots, M$ such that each of the components of $\boldsymbol{z}[t]$ is given by

$$z_m[t] = \begin{cases} \left|A\sin\left(\frac{M\pi t}{T}\right)\right|, & \text{if } t \in I_m \\ 0 & \text{otherwise} \end{cases} \qquad (14)$$

This signal is then projected onto the neurons through a decoder $\boldsymbol{\mu} \in \mathbb{R}^{N \times M}$ similar to $\eta$ in Eq. (10). The total current is then given by

$$I[t] = I_G[t] + I_Q[t] + \mu z[t] \qquad (15)$$

The aim of FORCE learning is to approximate a $K$-dimensional time-varying teaching signal $x[t]$. The vector $r[t]$ is used to obtain an approximant of the desired signal

$$\hat{x}[t] = \phi^T r[t] \qquad (16)$$

The weights are updated using the RLS learning rule according to:

$$\phi[t] = \phi[t - 1] - e[t]P[t]r[t] \qquad (17)$$

$$P[t] = P[t - 1] - \frac{P[t - 1]r[t]r[t]^T P[t - 1]}{1 + r[t]^T P[t - 1]r[t]} \qquad (18)$$

During the training phase, the teaching signal $x[t]$ is used to perform the RLS update. Then, during the testing phase, the teaching signal is removed.

We used the parameters given in Table 4 for all FORCE experiments, unless otherwise specified. For the gamma initialised synaptic time constants, we draw the values from $\Gamma(3, 0.025)$.

The period $T$ was chosen to be equal to the length of the teaching signal $x[t]$. The membrane potentials were randomly initialised following a uniform

**Table 4 FORCE network parameters.**

| Parameter | Value | Description |
|---|---|---|
| $\Delta t$ | 0.04 ms | Simulation time step |
| $N$ | 1000 | Number of neurons |
| $M$ | 500 | Number of HDTS subintervals |
| $\tau_m$ | 10 ms | Membrane time constant |
| $\tau_s$ | 20 ms | Synaptic time constant |
| $U_{th}$ | −40 mV | Membrane threshold |
| $U_0$ | 0 mV | Resting potential |
| $U_r$ | −65 mV | Reset potential |
| $t_{ref}$ | 2 ms | Refractory time |
| $Q$ | 10 | Chaotic weight |
| $G$ | 0.04 | Learned weight |
| $A$ | 80 | HDTS amplitude |

distribution $\mathcal{U}(U_r, U_{th})$. Vectors $\boldsymbol{\eta}$ and $\boldsymbol{\mu}$ were randomly drawn from $\mathcal{U}(-1, 1)$. The static weights $\nu$ were drawn from a normal distribution $\mathcal{N}(0, 1/(Np^2))$, then these weights were set to 0 with probability $p = 0.1$. All other variables are initialised to zero unless otherwise specified.

We computed the errors following

$$\text{logMSE}(\boldsymbol{x}, \hat{\boldsymbol{x}}) = \log_{10}\left(\frac{||\boldsymbol{x} - \hat{\boldsymbol{x}}||_F^2}{D}\right) \quad (19)$$

where $||\boldsymbol{x}||_F$ is the Frobenius norm of the multidimensional time signal $\boldsymbol{x}$ and $D$ is the number of elements in $\boldsymbol{x}$.

## Data availability

**Spiking datasets**

The spiking data used in this study are available in the following databases.
- N-MNIST: https://www.garrickorchard.com/datasets/n-mnist.
- Fashion-MNIST: https://github.com/zalandoresearch/fashion-mnist.
- DVS Gesture: https://www.research.ibm.com/dvsgesture.
- Heidelberg Spiking Datasets (SHD and SSC): https://compneuro.net/posts/2019-spiking-heidelberg-digits.

**Neural data**

The neural data used in this study are available in the following databases.
- Allen Atlas: https://allensdk.readthedocs.io/en/latest.
- Paul Manis dataset: https://figshare.com/articles/dataset/Raw_voltage_and_current_traces_for_current-voltage_IV_relationships_for_cochlear_nucleus_neurons_/8854352.

**Audio files**

The Zebra Finch bird song data used in this study is available in the following database.
- Zebra Finch bird song: https://www.ncbi.nlm.nih.gov/pmc/articles/PMC3192758/bin/pone.0025506.s002.wav.

## Code availability

All code is available at https://github.com/npvoid/neural_heterogeneity[35].

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

## Acknowledgements

We are immensely grateful to the Allen Institute and Paul Manis for publicly sharing their databases that allowed us to estimate time constant distributions in the brain. Releasing this data is not only immensely generous and essential for this work, but more generally it accelerates the pace of science and represents an optimistic vision of the future. The EPSRC Centre for Doctoral Training in High Performance Embedded and Distributed Systems (HiPEDS, Grant Reference EP/L016796/1) and the Imperial College President's PhD Scholarship is gratefully acknowledged.

## Author contributions

These authors contributed equally: N.P.-N. and V.C.H.L. N.P.-N.: conceptualisation, methodology, software, validation, visualisation, writing—original draft; V.C.H.L.: methodology, software, validation, visualisation, writing—review and editing; P.L.D.: writing—review and editing, supervision, project administration; D.F.M.G.: conceptualisation, resources, data curation, writing—review and editing, supervision, project administration.

## Competing interests

The authors declare no competing interests.
