## [Peer Review File · Nature Communications]

Reviewers' Comments:

Reviewer #1:

Remarks to the Author:

This paper discusses heterogeneous time constants in spiking neural networks. Both techniques, including heterogeneous initialization and trainable time constants, are compared and evaluated in tasks with varying degrees of temporal structure. The experimental results indicate that heterogeneity brings higher accuracy, better generalization performance, and less hyperparameter sensitivity (greater robustness).

Overall, I think this is a very well-written paper. I would appreciate it very much if the authors could comment on the following concerns.

1. The SNN used for SHD and SSC is claimed to be "a detailed model of the activity of bushy cells in the cochlear nucleus", which is not clear to me. I suggest the author should clarify the structure of this model.
2. Do other kinds of initialization (uniform, normal, etc.) of time constant lead to the same final distribution after heterogeneous training? Since only Gamma, Bernoulli, and homogeneous configurations are tested.
3. How the lower bound $e^{-1/3}$ in the clipping function of time constants is chosen? Will it affect the final distribution?
4. A statistical hypothesis testing could be more persuasive for the observed coherence between the final distribution and in vivo data.
5. The terminology "robustness" is easily confused with robustness towards adversarial attack in the machine learning context. Does "sensitivity to hyperparameters" convey your ideas better?
6. Does heterogeneous training alone improve generalization performance on the augmented SHD spoken digits datasets?

Reviewer #2:

None

Reviewer #3:

Remarks to the Author:

What are the major claims of the paper?

The major claims of the paper are that introducing heterogeneity in neuronal time constants, either through initialization or through learning, improves the task performance in particular tasks with a high temporal complexity: it increases the speed of learning (i.e. how many epochs are needed), the final accuracy, the generalisability and the robustness against mismatch in the hyperparameters.

Are they novel and will they be of interest to others in the community and the wider field? Is the work convincing, and if not, what further evidence would be required to strengthen the conclusions?

The importance of neural heterogeneity for information representation and transfer has been commented on before, although not by many. However, the fact that this is explicitly calculated as task performance and investigated using both different tasks and learning algorithms is novel and convincing.

On a more subjective note, do you feel that the paper will influence thinking in the field?

Yes, I believe so. Although neural heterogeneity has been commented on before, as noted above, has often been ignored by the (computational) neuroscience community at large, except for a few researchers. This convincing article will help show that this should no longer be the case.

I believe that this is a convincing paper, that makes a relevant case. However, some choices and explanations, especially in the Materials and Methods section, could be explained better. Also, the 'readme' of the code only has a list of links to the datasets, no explanation on how to use the code or what roughly the structure of the code is. Without such an explanation, it is quite difficult to

review the code. The FORCE code does not have a 'readme'. So I believe the code should contain more explanation.

Specific comments:

Materials and Methods

I believe the Methods section needs some improvement, especially on some of the choices that were made:

In line 350 and equation (2), a spike train of layer l is defined as $l-1$ (in the equation).

The discretisation on p. 16 (eq 5 and 6) is not very standard (it is not a standard Euler or Runge-Kutta), with the exponential values α and β . While that is ok (I can see that it would work), I would like to hear some motivation about this: why is this particular discretization used? What is it based on?

The notation of time indices versus time values is confusing: t is normally used for time values, but the ' $t+1$ ' in equations 5 and 6 suggest it is a time index rather than a time value (otherwise it would have stated ' $t + \Delta t$ '). In the FORCE training it is even more confusing: ' $t+1$ ' is used in equations 11-13, but ' $t + \Delta t$ ' in equation 17.

In line 384 abbreviations SNN and RNN BPTT are used, but not defined.

I find the clipping function (line 396) confusing: how were these values chosen and why? What are these numbers based on?

In line 414, the loss function is defined as max over time. Why was this loss function chosen (and not for instance mean over time, or another loss function)?

In line 423 it is stated that a log-normal distribution was used, but not for what it was used.

Similarly, it is not explained why in line 427/428 5% of the neurons were given a time constant of 100ms.

Was there no separation of training and test data for the surrogate gradient descent training? If so, why not, is there no overfitting? If not so, please explain how this separation was made.

Tables 2,3 and 4 could use an extra column explaining what the parameters are (i.e. a description of the parameter)

On p. 19: The Nicola and Clopath paper (reference [13]) has many examples. Please explain what example was followed here, as it is not clear and quite hard to deduce.

Line 451: neurons leaving an equation actually sounds quite funny.

Please define $r(t)$ in equation 16-18: it is needed to update the weights ϕ , but it is not stated how it is set.

Other Comments

Also in the general text, some choices and reasoning could be explained a bit better:

In line 40, a few words could be spent on that 'heterogeneous learning' is actually learning of time constants (as this is not stated) and how these time constants are learnt, as this is not a standard procedure.

In line 77, it is not quite clear how this 0.23% is calculated, could this be explained (is it the # neurons / # synapses?). And could this be quantified using for instance the Akaike information criterion or the Bayesian information criterion or something similar (see Pitt, M. a., & Myung, I. J. (2002). When a good fit can be bad. *Trends in Cognitive Sciences*, 6(10), 421–425.

[https://doi.org/10.1016/S1364-6613\(02\)01964-2](https://doi.org/10.1016/S1364-6613(02)01964-2)?

In line 80 'adding heterogeneity'-- which one? The initialization or the learning heterogeneity?

Also line 80/81: explain why adding heterogeneity is $O(n)$ and adding more neurons is $O(n^2)$.

Line 82: I don't understand the point about BrainScaleS. Why is this mentioned?

I think actually line 80-83 should be moved to the Discussion section.

Line 99: This paper also shows an adaptive threshold leads to better performance: Huang, C., Resnik, A., Celikel, T., & Englitz, B. (2016). Adaptive Spike Threshold Enables Robust and Temporally Precise Neuronal Encoding. *PLoS Computational Biology*, 12(6).

<https://doi.org/10.1371/journal.pcbi.1004984>

Figure 2:

In A: are in the DVS, SHD and SSC case the SEM values so small that they are invisible? Please mention that in that case.

In C: Where do these data come from? Please put proper references in caption.

In E and F: I find label 'Scale' a bit too general: could it be called 'time scale factor' or something

similar?

Line 107: 'similar distributions': Similar how? In what aspect? Please explain.

Line 113: I think the point of this paragraph is that there is better generalization. So could the title be something like: 'Heterogeneity improves generalizability: speech learning across time scales'.

Line 114: 'Sensory signals (...) can be recognized (...)' Recognized by whom? Neural networks? Humans? Animals?

Line 120-126: I don't understand these sentences. What do you mean by 'generalize better to time scales outside the training distributions'? Why would chance performance be 5%? Or is this the case for the whole SHD dataset? I find again 'time scale' a bit confusing, something like 'time multiplication factor' or something would be clearer. And I don't understand the last sentence: how was the distribution fine tuned? What figure panel was this based on and how?

Line 138-140: please explain at what figure panel this can be seen and how.

Line 152: please give the parameters of the gamma distribution (or a reference to where I can find them)

Why did you only look at network size, and then mostly at a larger network size, as 'mistuning', and not other parameters? I suppose the network size is kind of like a proxy to have 'similarly mistuned' G and Q, but I don't completely see the reasoning behind this. Please explain why this parameter was chosen!

Line 160: What becomes saturated? Do you mean the neurons fire at their maximal frequency? Or the synapses become saturated?

Line 175-177: I don't understand what you mean with this comment or why it is there.

Figure 3

How was the error calculated? At the raw signal or the spectrogram? Was it normalized in some way?

A: it is not a spectrogram of a zebra finch, but of a zebra finch call

B: please give the optimal hyperparameters mentioned. It is impossible to see from D, as basically everything is dark blue.

D: It appears that the network of 500 neurons actually performs better than the 1000 neuron network, even though the 'correct' one is 1000 neurons, how is this possible?

Line 182/183: like in line 80/81, please explain how this $O(n)$ and $O(n^2)$ is deduced.

Line 188/189: there is no question that remains? What is the question?

Line 192/193: I don't understand this sentence.

Line 195: what does 'these brain regions' refer to? I see no brain regions mentioned before.

Line 208: I thought Figure 3 contained thousands, not hundreds of neurons?

Line 211: 'Beyond this' is not the start of a new paragraph

The attached pdf shows some minor text edits.

We thank the reviewers for their careful reading of our paper and their comments. We have conducted some new simulations and made a number of modifications to the text based on their comments. Below, we have included the reviews and our line by line replies. Parts of the reviews that did not need a response are in grey, and our responses are in blue.

Reviewer #1 (Remarks to the Author):

This paper discusses heterogeneous time constants in spiking neural networks. Both techniques, including heterogeneous initialization and trainable time constants, are compared and evaluated in tasks with varying degrees of temporal structure. The experimental results indicate that heterogeneity brings higher accuracy, better generalization performance, and less hyperparameter sensitivity (greater robustness).

Overall, I think this is a very well-written paper. I would appreciate it very much if the authors could comment on the following concerns.

1. The SNN used for SHD and SSC is claimed to be “a detailed model of the activity of bushy cells in the cochlear nucleus”, which is not clear to me. I suggest the author should clarify the structure of this model.

We added some further clarification of the structure of this model, which was developed by Cramer et al. (2020).

2. Do other kinds of initialization (uniform, normal, etc.) of time constant lead to the same final distribution after heterogeneous training? Since only Gamma, Bernoulli, and homogeneous configurations are tested.

We ran additional simulations with others initialisations: uniform distribution, normal distribution and homogeneous at 100ms. We show that these distributions end in similar final distributions after testing despite starting at very different initialisations. We included these figures in the newly added Supplementary Materials S9 and S10.

3. How the lower bound $e^{-1/3}$ in the clipping function of time constants is chosen? Will it affect the final distribution?

This lower bound corresponds to a time constant $\tau = 3\Delta t = 1.5\text{ms}$, where Δt is the time step of the simulation. Time constants that are too small compared to the time step can lead to numerical instability, and experimentally we found that $3\Delta t$ was a reasonable cut-off to avoid this. Making the lower bound any lower results in some neurons getting stuck in a constant firing regime where no learning could take place.

As for the upper bound. We chose $\tau_{\text{max}} = 100\text{ms}$ informed by the time constants reported in the NeuroElectro database (https://neuroelectro.org/ephys_prop/4/) which shows that about 99.5% of the membrane time constants recorded from biological neurons have time constants below 100ms. Since there is no data available on synaptic time constants we chose the same range as the membrane time constants.

We added this clarification in the methods.

4. A statistical hypothesis testing could be more persuasive for the observed coherence between the final distribution and in vivo data.

We initially attempted to do this, but it was unclear to us whether or not the analysis was meaningful. Our model distributions almost but do not perfectly fit gamma and/or log-normal distributions (which seemed like natural choices) using a chi-squared goodness of fit test. The same is true for the experimental data from Manis and the Allen institute. In some cases, gamma and/or log-normal gave a good fit, and in other cases it did not. There are a number of possible reasons for this.

One reason could be that there is a different 'true' distribution that is similar to both gamma and log-normal, but then we don't have any strong argument as to what this should be.

Another possibility is that each cell type may have its own distinct distribution, and that the experimental data includes cells of different types. We have some experimental support for this in the sense that when we lumped all data together we could clearly see multiple peaks in the distributions, suggesting a mixture of several distributions with different parameters. We did our best to separate cell types based on the information available in the Allen dataset, dividing into putative excitatory and inhibitory cells, and restricting to a single area. In most cases, the observed distributions after this separation seem to have a single or at least smaller number of peaks. However, at this point the number of cells broken down into these increasingly smaller classes makes it difficult to make a strong conclusion on this point.

There is also an issue with the model data in that we artificially clamp time constants between a minimum and maximum value, and in particular we see a clear peak at the maximum values suggesting that the clipping had an effect. It might be possible to correct for this in the statistical test, but the mere act of clamping those values might change the rest of the distribution slightly (because the optimal distribution with this restriction may be slightly different to the optimal distribution without this restriction). So it is again difficult to make a clear argument that the results do or do not come from any particular distribution.

An alternative approach would be to directly compare the model and experimental histograms in a distribution-free way, however we do not see a way to do this given that we do not expect our model results to directly match any particular area or cell type in any particular species. We might expect different means, standard deviations, shape parameters, etc.

This final point is perhaps the clearest: we trained our model on a task that - despite being much more detailed and 'realistic' in some ways - still doesn't match the complexity of real environments, nor the range of tasks that the animal has to carry out. So we wouldn't expect the two distributions to be identical. We found it striking that they were as similar as they were given the simplicity of the task.

For all the above reasons, we decided that the fairest and most reasonable statement we could make was to observe that the distributions are "similar" while avoiding making a specific claim about what the distribution is and whether or not it was exactly the same. We have updated the text to more explicitly state that we would not expect a perfect match but that the similarity is striking given the relative simplicity of the task. We also added some model fit curves to the supplementary materials (S11-13).

5. The terminology "robustness" is easily confused with robustness towards adversarial attack in the machine learning context. Does "sensitivity to hyperparameters" convey your ideas better?

We think that this paper is likely to be read primarily from a neuroscience perspective, as the use of spiking neural networks is not yet widespread in the machine learning community. In this context, we believe robustness would generally be understood to mean robust learning in the face of new environments, perturbations, etc. which is how we use it here (first used in the section on "Heterogeneity improves robustness against mistuned learning" where it is clearly used in this sense).

6. Does heterogeneous training alone improve generalization performance on the augmented SHD spoken digits datasets?

We were not entirely sure what this question refers to, and we have therefore attempted to address both the interpretations we could think of.

If the question refers to our augmented SHD dataset as introduced in lines 122-124 of the original submission, then looking at figure 2E we can see that heterogeneous training alone does indeed improve performance at all Time scale factors. This corresponds to the green line (heterogeneous training alone) against the blue line (homogeneous initialisation and training).

Alternatively, the question may be referring to other SHD data augmentations present in the literature such as in Cramer et al. (2020). To check this, we trained networks following the data augmentations referred in that paper under a fully homogeneous configuration and under a heterogeneous training configuration with homogeneous initialisation. We added these results to the newly added Supplementary Materials S14. The results obtained show that while an improvement in performance is present in both configurations, the heterogeneous network still performs better.

Reviewer #3 (Remarks to the Author):

What are the major claims of the paper?

The major claims of the paper are that introducing heterogeneity in neuronal time constants, either through initialization or through learning, improves the task performance in particular tasks with a high temporal complexity: it increases the speed of learning (i.e. how many epochs are needed), the final accuracy, the generalisability and the robustness against mismatch in the hyperparameters.

Are they novel and will they be of interest to others in the community and the wider field? Is the work convincing, and if not, what further evidence would be required to strengthen the conclusions?

The importance of neural heterogeneity for information representation and transfer has been commented on before, although not by many. However, the fact that this is explicitly calculated as task performance and investigated using both different tasks and learning algorithms is novel and convincing.

On a more subjective note, do you feel that the paper will influence thinking in the field?

Yes, I believe so. Although neural heterogeneity has been commented on before, as noted above, has often been ignored by the (computational) neuroscience community at large, except for a few researchers. This convincing article will help show that this should no longer be the case.

I believe that this is a convincing paper, that makes a relevant case. However, some choices and explanations, especially in the Materials and Methods section, could be explained better. Also, the 'readme' of the code only has a list of links to the datasets, no explanation on how to use the code or what roughly the structure of the code is. Without such an explanation, it is quite difficult to review the code. The FORCE code does not have a 'readme'. So I believe the code should contain more explanation.

Many thanks for mentioning this point. We have updated the code with a detailed readme file that we hope should make it clear how to run it.

Specific comments:

Materials and Methods

I believe the Methods section needs some improvement, especially on some of the choices that were made:

In line 350 and equation (2), a spike train of layer l is defined as $l-1$ (in the equation).

We corrected the layer indices to be consistent across equations.

The discretisation on p. 16 (eq 5 and 6) is not very standard (it is not a standard Euler or Runge-Kutta), with the exponential values α and β . While that is ok (I can see that it would work), I would like to hear some motivation about this: why is this particular discretization used? What is it based on?

On further inspection, we agree that the discretisation is unusual. We used it because it is standard in the comparable literature (Cramer et al. (2020), Bellec et al. (2019), and Yin et al. (2020)). None of these papers give a derivation of this approximation, but it can be found by assuming that $l(t)$ is constant over the interval $(t, t+dt)$ and then solving analytically. However, you could solve the pair of

equations for u and I analytically without making this assumption and get a slightly more accurate solution. However, this change would make almost no overall difference, for the following reason.

There is already an error of order $O(dt)$ due to forcing spike times to align to the grid, and this cannot be avoided in this framework. The difference between the solution assuming $I(t)$ is constant over the interval and the more accurate one is $O(dt^2)$, so using the more accurate approach doesn't change the order of the overall error, which will remain $O(dt)$. Using Euler or a higher order Runge-Kutta method would also not improve the order of the error which will always be dominated by the $O(dt)$ term introduced by forcing spikes to align to the grid.

We modified the text to make it clear that we use a standard discretisation from the literature.

The notation of time indices versus time values is confusing: t is normally used for time values, but the ' $t+1$ ' in equations 5 and 6 suggest it is a time index rather than a time value (otherwise it would have stated ' $t + \Delta t$ '). In the FORCE training it is even more confusing: ' $t+1$ ' is used in equations 11-13, but ' $t + \Delta t$ ' in equation 17.

We added an explanation in the text to clarify the use of square brackets to index variables changing in a discrete time setting. We also modified equations 17 and 18 to be consistent with the other equations.

In line 384 abbreviations SNN and RNN BPTT are used, but not defined.

We added definitions.

I find the clipping function(line 396) confusing: how were these values chosen and why? What are these numbers based on?

The lower bound corresponds to a time constant $\tau=3dt=1.5ms$, where dt is the time step of the simulation. Time constants that are too small compared to the time step can lead to numerical instability, and experimentally we found that $3dt$ was a reasonable cut-off to avoid this. Making the lower bound any lower results in some neurons getting stuck in a constant firing regime where no learning could take place.

As for the upper bound. We chose $\tau_{max}= 100ms$ informed by the time constants reported in NeuroElectro database (https://neuroelectro.org/ephys_prop/4/) which shows that about 99.5% of the membrane time constants recorded from biological neurons have time constants with values below 100ms. Since there is no data available on synaptic time constants we chose the same range for as for the membrane constant.

We added this clarification in the methods.

In line 414, the loss function is defined as max over time. Why was this loss function chosen (and not for instance mean over time, or another loss function)?

We followed the decision in Cramer et al. (2020), both to be comparable to existing literature and also because they found that this choice of loss function led to the highest classification performance.

In line 423 it is stated that a log-normal distribution was used, but not for what it was used.

It is a log-normal distribution for the initialisation of the time constants. We have clarified this in the text.

Similarly, it is not explained why in line 427/428 5% of the neurons were given a time constant of 100ms.

We wanted to test whether or not a longer "memory" was all that was needed, and when training time constants we found that a small proportion would reach these maximum values. We have added this to the text.

Was there no separation of training and test data for the surrogate gradient descent training? If so, why not, is there no overfitting? If not so, please explain how this separation was made.

We did separate the training samples into training and testing for all datasets. In all cases we used the train/test split suggested by the corresponding dataset authors as to be able to compare with other results in the literature that used the same split. There is indeed some overfitting, however the results reported in Table 1 and Figures 2 and 3 correspond only to testing accuracies. We added a clarification on how the split was made on the text as well and we made clear that results in the figures correspond to testing accuracies.

Tables 2,3 and 4 could use an extra column explaining what the parameters are (i.e. a description of the parameter)

We added this description in the tables.

On p. 19: The Nicola and Clopath paper (reference [13]) has many examples. Please explain what example was followed here, as it is not clear and quite hard to deduce.

We added this.

Line 451: neurons leaving an equation actually sounds quite funny.

We rephrased this.

Please define $r(t)$ in equation 16-18: it is needed to update the weights ϕ , but it is not stated how it is set.

We defined $r[t]$ in equation (13), although we had missed out the initial condition $r[0]=0$. We have now added this in the text.

Other Comments

Also in the general text, some choices and reasoning could be explained a bit better:

In line 40, a few words could be spent on that 'heterogeneous learning' is actually learning of time constants (as this is not stated) and how these time constants are learnt, as this is not a standard procedure.

We have made this clearer in the revised version.

In line 77, it is not quite clear how this 0.23% is calculated, could this be explained (is it the # neurons / # synapses?). And could this be quantified using for instance the Akaike information criterion or the Bayesian information criterion or something similar (see Pitt, M. a., & Myung, I. J. (2002). When a good fit can be bad. Trends in Cognitive Sciences, 6(10), 421–425. [https://doi.org/10.1016/S1364-6613\(02\)01964-2](https://doi.org/10.1016/S1364-6613(02)01964-2))?

That's correct, it's actually twice the number of neurons (because we added two neuron-specific time constants) / number of synapses. This particular value was calculated on for the SHD dataset and 128 hidden neurons (#synapses=700x128+128x128+20x128, #hidden neurons x2 = 128x2). We added some text to clarify this. We don't think that computing a Bayes or Akaike information criterion is necessary or appropriate here as they are designed to prevent overfitting when fitting statistical models to data, but we test on held out data instead.

In line 80 'adding heterogeneity'-- which one? The initialization or the learning heterogeneity?

We mean that modifying the neuron model to make time constants heterogeneous adds complexity (memory and computation) at inference time (storing and fetching the time constants). We updated the text to reflect this.

Also line 80/81: explain why adding heterogeneity is $O(n)$ and adding more neurons is $O(n^2)$.

We added this.

Line 82: I don't understand the point about BrainScaleS. Why is this mentioned?
I think actually line 80-83 should be moved to the Discussion section.

We agree that it would fit better in the discussion and we moved it there.

Line 99: This paper also shows an adaptive threshold leads to better performance: Huang, C., Resnik, A., Celikel, T., & Englitz, B. (2016). Adaptive Spike Threshold Enables Robust and Temporally Precise Neuronal Encoding. PLoS Computational Biology, 12(6). <https://doi.org/10.1371/journal.pcbi.1004984>

We added this citation.

Figure 2:

In A: are in the DVS, SHD and SSC case the SEM values so small that they are invisible? Please mention that in that case.

Yes, this is the case. We added a mention of this in the text.

In C: Where do these data come from? Please put proper references in caption.

We added this.

In E and F: I find label 'Scale' a bit too general: could it be called 'time scale factor' or something similar?

We updated this.

Line 107: 'similar distributions': Similar how? In what aspect? Please explain.

Both the model fits and experimental data are almost but not quite log-normal or gamma distributed (see new supplementary materials S11-13). However, since they are not perfect fits for these distributions we were not able to make this "similarity" more precise. See our reply to reviewer 1 for a more detailed response on this point.

Line 113: I think the point of this paragraph is that there is better generalization. So could the title be something like: 'Heterogeneity improves generalizability: speech learning across time scales'.

We made this change.

Line 114: 'Sensory signals (...) can be recognized (...)' Recognized by whom? Neural networks? Humans? Animals?

We added "by humans or animals" to clarify this.

Line 120-126: I don't understand these sentences. What do you mean by 'generalize better to time scales outside the training distributions'? Why would chance performance be 5%? Or is this the case for the whole SHD dataset? I find again 'time scale' a bit confusing, something like 'time multiplication factor' or something would be clearer. And I don't understand the last sentence: how was the distribution fine tuned? What figure panel was this based on and how?

During training we sampled the scaling factor from the distribution in grey in figure 2E. With this distribution the network was never shown samples at scales >3 . Then on testing we uniformly draw scales in the interval $[0.2, 4]$. The results show that the heterogeneous regimes perform better at out-of-sample scales, thus they generalise better.

The chance performance for the SHD dataset is 5% because there are 20 classes.

By fine tuning the distribution we simply mean training the time constants. This corresponds to figure 2E. We agree that the wording is confusing and we rephrased it.

Line 138-140: please explain at what figure panel this can be seen and how.

We added this.

Line 152: please give the parameters of the gamma distribution (or a reference to where I can find them)

They are drawn from $\text{Gamma}(3, 0.025)$. We added this to the methods.

Why did you only look at network size, and then mostly at a larger network size, as 'mistuning', and not other parameters? I suppose the network size is kind of like a proxy to have 'similarly mistuned' G and Q, but I don't completely see the reasoning behind this. Please explain why this parameter was chosen!

We consider three parameters only, network size N, G and Q. In Fig 3D we vary all of these parameters, while in Fig 3B we vary only N. You can think of Fig 3B as a slice of Fig 3D. We kept the panel 3B because it is slightly easier to read than 3D and because it contains 10 trials on these values to demonstrate consistency across trials. We added some text to the caption to make this clear.

Line 160: What becomes saturated? Do you mean the neurons fire at their maximal frequency? Or the synapses become saturated?

Yes, it means the neurons are firing at or close to their maximal rate. We updated the text to clarify this.

Line 175-177: I don't understand what you mean with this comment or why it is there.

We meant to say that the optimal time constant distributions in real animals could have been found by evolution, and do not need to be learned during an animal's lifetime (although we also allow for that possibility). We have rephrased this sentence to make it clearer.

Figure 3

How was the error calculated? At the raw signal or the spectrogram? Was it normalized in some way?

The error was calculated at the raw signal as the log MSE of the signal. We added the exact expression in the methods.

A: it is not a spectrogram of a zebra finch, but of a zebra finch call

We updated this.

B: please give the optimal hyperparameters mentioned. It is impossible to see from D, as basically everything is dark blue.

We added this.

D: It appears that the network of 500 neurons actually performs better than the 1000 neuron network, even though the 'correct' one is 1000 neurons, how is this possible?

The simulations in this figure (3D) are not optimised for any particular number of neurons. Here we only want to study the change on the area of stability as we vary the number of neurons and the hyperparameters G and Q for each configuration. What we find is that areas of stability tend to be larger for smaller number of neurons and, more importantly, heterogeneous distributions.

We used the optimal Q and G values found in 3D for N=1000 neurons to produce 3B on 10 trials. In 3B, the error values obtained for 1000 neurons are still below those obtained for 500 although is difficult to see as they are very close.

Line 182/183: like in line 80/81, please explain how this $O(n)$ and $O(n^2)$ is deduced.

We added an explanation in the Results section where these complexities were introduced for the first time.

Line 188/189: there is no question that remains? What is the question?

We have reworded this.

Line 192/193: I don't understand this sentence.

We have reworded this.

Line 195: what does 'these brain regions' refer to? I see no brain regions mentioned before.

We meant that it is hard to directly compare our simulation results to time constant distributions observed in particular species and brain regions, because the circuitry of those regions is different, and they are not solving the same task. We have reworded this to make it clearer.

Line 208: I thought Figure 3 contained thousands, not hundreds of neurons?

We clarified this by noting that we are limited to hundreds when using surrogate gradient descent and thousands using FORCE training

Line 211: 'Beyond this' is not the start of a new paragraph

We fixed this.

The attached pdf shows some minor text edits.

We have updated the text.

Reviewers' Comments:

Reviewer #1:

Remarks to the Author:

All my concerns have been addressed in this revised manuscript. I have no further comments.

Reviewer #3:

Remarks to the Author:

I have no further comments, the authors have addressed all my points satisfactory.